# Identification of Highlighted Cells in Low-Variance Raster Data Application to Digital Elevation Models

**Manuel Antonio Ureña-Cámara**  **and Antonio Tomás Mozas-Calvache \*** 

Departamento de Ingeniería Cartográfica, Geodésica y Fotogrametría, University of Jaén, 23071 Jaén, Spain; maurena@ujaen.es
\* Correspondence: antmozas@ujaen.es; Tel.: +34-953-212713

**Abstract:** This study describes a new algorithm developed to detect local cells of minimum or maximum heights in grid Digital Elevation Models (DEMs). DEMs have a low variance in digital levels due to the spatial continuity of the data. Traditional algorithms, such as SIFT, are based on statistical variance, which present issues to determine these highlighted cells. However, one of the main purposes of this identification is the use of these points (cells) to assess the positional accuracy of these products by comparing those extracted from the DEM with those obtained from a more accurate source. In this sense, we developed an algorithm based on a moveable window composed of variable sizes, which is displaced along the image to characterize each set of cells. The determination of highlighted cells is based on the absolute differences of digital levels in the same DEM and compared to those obtained from other DEMs. The application has been carried out using a great number of data, considering four zones, two spatial resolutions, and different definitions of height surfaces. The results have demonstrated the feasibility of the algorithm for the identification of these cells. Thus, this approach expects an improvement in traditional procedures. The algorithm can be used to contrast DEMs obtained from different sources or DEMs from the same source that have been affected by generalization procedures.

**Keywords:** image matching; low-variance feature detection; DEM matching; DEM quality control

## 1. Introduction

Digital Elevation Models (DEMs) are widely used to represent terrain topography (Digital Terrain Models (DTMs)) and other surfaces (Digital Surface Models (DSMs)), including those features related to the biosphere, anthroposphere, etc. [1]. From a geomatic point of view, DEMs represent height values using raster and vector models. Raster DEMs are based on grids, composed of a regular array of cells, including heights, as an attribute (this model is commonly referred to as 2.5D data because it only supports a single z-value for each planimetric location in a specific column and row), while vector DEMs are based on 3D surfaces composed of meshes derived from a set of 3D points. The raster model considered in this study can be managed using images that represent height as digital levels of pixels determined by a given bit-depth capacity (e.g., ArcGIS Pro uses 32-bit depth in grids [2]. These values can be obtained from a single sample point or the average measurements representing the height values of a given area (e.g., the area covered by the pixel). In any case, the values of adjacent cells are highly correlated because these surfaces represent elements with a physical continuity (e.g., terrain), where drastic changes are not common (except in cases of steep slopes, such as gorges, cliffs, etc.). Therefore, we must consider a low variance in this data, at least in a close environment. This aspect is very important when we are detecting points (e.g., using point detection algorithms) in grid DEMs because the location of a specific point can be influenced by its surroundings.

The height values contained in DEMs are derived from measurements that are characterized by a certain level of accuracy that is influenced by acquisition and processing errors.

The ISO 19157 standard [3,4] defines five elements of data quality, including positional accuracy, which is related to the location of features. In this context, positional accuracy has three sub-elements: absolute or external accuracy, relative or internal accuracy, and gridded data position accuracy. These are related to the closeness of the reported coordinate values, relative positions, and gridded data positions to values accepted as true, respectively [3]. This aspect implies that the assessment of the positional accuracy of any cartographic product must include a comparison of data (absolute or relative positions), with respect to other independent datasets that are considered true values. The accuracy of these reference datasets should be at least three times higher than that of the product to be assessed [5]. In the case of DEMs, positional accuracy has been widely analyzed in the literature [6], considering error causes and consequences and the methodologies developed to assess them [7,8], by considering several aspects (data model, data acquisition, accuracy measures, etc.). In addition, some standards [9–12], published during the last decade by several institutions, have described the procedures to carry out this assessment. These studies have traditionally considered vertical accuracy without considering the effect of horizontal accuracy. It is unusual to include a report about horizontal accuracy [6,7], although there are some studies that have analyzed the impact of horizontal errors on the accuracy of DEMs, proposing an improvement in the final vertical accuracy after correcting them [13].

Although there are recent initiatives proposing the use of surfaces to check DEMs [14], most of the studies developed until now are based on a comparison of a set of checkpoints, whose coordinates are obtained from a more accurate independent source, and using some measures (e.g., Root Mean Squared Error (RMSE)) to describe the deviations with respect to the value that is considered true. These checkpoints are usually well-defined points that are selected and measured manually. As an example, the ASPRS standard [12] indicates that well-defined points should be easily visible or recoverable on the ground, on the independent source of higher accuracy, and on the product itself. This supposes that checkpoints should contain certain characteristics that make them geometrically particular, avoiding other points that are not so easily identifiable. Most of the studies that have analyzed the accuracy of DEMs used checkpoints based on Ground Control Points (GCPs), which are obtained in the field using surveying techniques, such as Global Navigation Satellite Systems (GNSS). The use of GCPs influences their distribution because they are usually located in accessible areas in order to reduce field costs. To avoid the use of GCPs, this study focuses on the use of another, more accurate DEM as a reference. The use of this reference adds another issue related to the differences in the spatial resolutions between the reference DEM and the destination DEMs and their repercussions on the identification of checkpoints.

The manual selection and measurement of checkpoints supposes a great deal of time and cost effort for any affected project, depending on, for example, the sample size of the points to be used. In addition, this manual selection is conditioned by the operator's experience. We must consider that two different operators will select different checkpoints following their own criteria, causing a certain bias. The use of an automatic procedure will avoid this problem because the selection will follow certain criteria, giving the process greater repeatability.

Considering the aspects described previously, an automatic procedure to determine checkpoints from both sources (to be assessed and referenced) is desirable. The use of an automatic procedure will provide a cost reduction and an improvement in the assessment due to the increase in the sample size by covering the DEM with a larger density of checkpoints. In addition, the automatization would remove the biases caused by operators. On the other hand, this assessment should analyze both vertical and horizontal accuracies in order to study the influence of the horizontal discrepancies in height values. In this study, we propose the possibility of using a set of checkpoints obtained directly from both sources and, more specifically, from two DEMs (the one to be assessed and the reference one). To achieve this objective, it is necessary to develop a new methodology that allows the automatic detection of homologous points from both sources but also considers the

characteristics of DEMs (spatial continuity, low variance, etc.) and the necessity of using well-defined points. As mentioned previously, the ASPRS standard [12] establishes some conditions applicable to checkpoints to be used in the positional accuracy assessment. These checkpoints must be clearly identifiable, both vertically and horizontally. Therefore, these aspects must be taken into account when developing this automatic procedure.

The automatic detection of homologous points contained in several images has been extensively studied during the last few decades. Several procedures based on point detectors have been described for their use in computer vision and photogrammetry, such as the Scale Invariant Feature Transform (SIFT) [15] and the Speeded Up Robust Features (SURF) [16]. Some recent studies have compared these detectors [17,18], and some applications have been performed with DEMs based on these algorithms, such as those oriented to the registration of several datasets [19,20]. Although these algorithms are focused on obtaining interest points that are common to several images, and these homologous points could be used to compare their coordinates, most of them are not suitable for a positional accuracy assessment because they cannot be assumed as well-defined points (considering the conditions indicated by the standards). Obviously, some of them could meet the requirements. This implies that a filtering procedure should be included to discriminate between those well-defined points, but this suggests an increase in the computing time. In this sense, a strategy based on the detection of highlighted points carried out from the beginning of the procedure, aimed at obtaining well-defined points, should be more effective.

### 1.1. Statistical Characterization of DEMs

One of the assumptions of this study is the low variance of the data represented in DEMs, and as a consequence, there is a need to develop a new methodology that considers this aspect to detect the highlighted points to be used in accuracy assessment procedures. To contrast this assumption, we have developed a preliminary analysis comparing the statistics of several images representing DEMs with other real images, such as those commonly used with the SIFT algorithm. In this regard, we used 12 official DEMs related to 3 cartographic sheets obtained from an official institution in Spain, which are more specifically described in Section 3 of this document, and 904 images obtained from the Image Matching Challenge PhotoTourism (IMC-PT) 2020 dataset [21]. These images are shared to both train and evaluate new strategies for image matching and, more specifically, to obtain dense and accurate 3D reconstructions from large collections of images based on the SIFT algorithm. Considering these datasets, we developed a statistical description based on the normalized values of the mean and the standard deviation of the digital levels of each image. The results are shown in Figure 1, where the graph displayed in Figure 1a shows the statistics of the mean values, and Figure 1b shows those related to the standard deviations. In the case of IMC-PT, the median value of all the means is close to 0.5 (Figure 1a), while the average standard deviation is higher than 0.3 (Figure 1b). On the other hand, the DEMs show a similar median value of the means of all the images with respect to the other dataset (Figure 1a), although this value is highly conditioned by the relief of the zones analyzed. However, the average standard deviation is lower than 0.2 and shows less variability (Figure 1b). This reduced data dispersion confirms that DEMs have a lower variance than the IMC-PT images analyzed and a lack of outlier values. Consequently, this aspect should be considered in point detection procedures because of its possible influence on feature identification. Interestingly, this lack of outliers indicates that an algorithm based on variance should have poor results and low highlighted point identification.

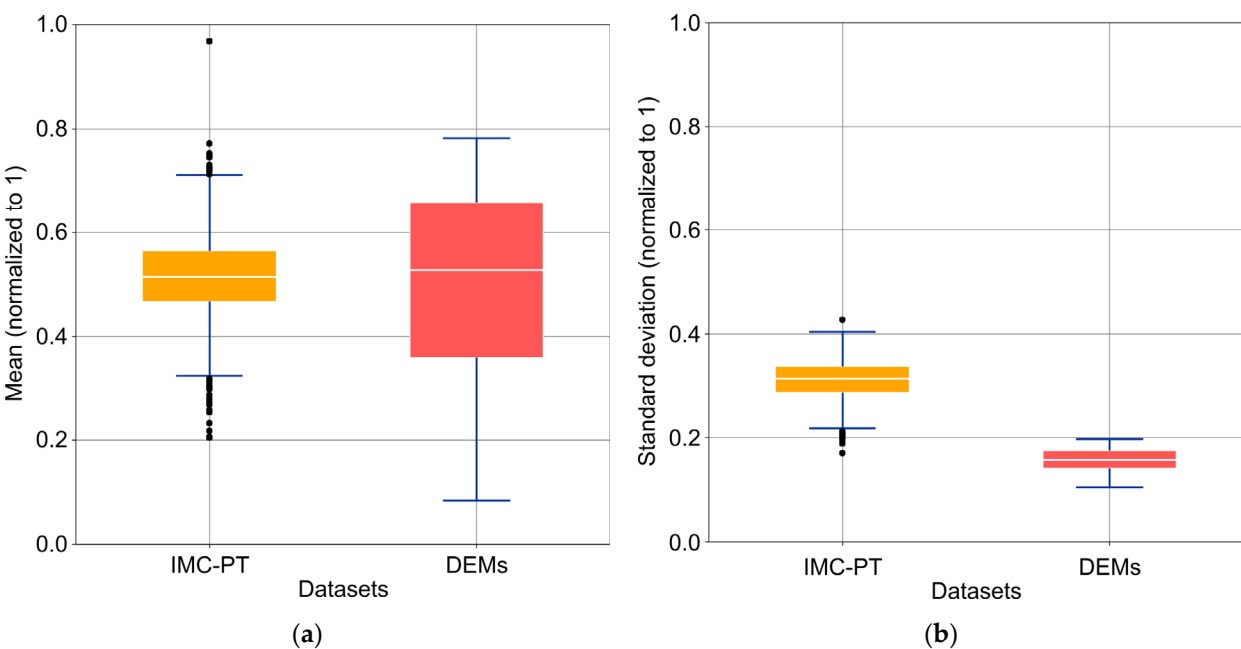

**Figure 1.** Statistics of datasets IMC-PT and DEMs: (**a**) mean values; (**b**) standard deviations.

### 1.2. Objectives of This Study

Taking into account the aspects described previously, the main objective of this study involves the development of a new methodology to automatically detect the homologous points from two DEMs (to be assessed and reference) that are suitable to assess the positional accuracy (vertical and horizontal) of the dataset to be analyzed. This implies that points must be well-defined, both on the sources and on the ground (following the ASPRS standard [12]). As a secondary goal, the procedure should allow us to increase the sample size of the points, improving the results given by the manual procedures. Another goal is related to the analysis of those parameters needed to execute the procedure and their relationships with the data characteristics, such as the spatial resolution of both DEMs.

This paper is structured as follows: First, a description of the proposed methodology is outlined, including the algorithm used to achieve the main objective of this study. After that, a description of the data used in the application of this methodology is given. Subsequently, a summary of the main results obtained after this application is given, including a discussion of the results and their impact on quality assessment, considering those aspects that affect the proposed methodology. Finally, the main conclusions of this study and future works will be presented.

### 2. Methodology

The methodology is designed to face the two main issues in DEMs. The first issue is the low variance mentioned in the previous section. The second issue is a general difference, both in scale and position. The latter requires further explanation. Different DEMs may describe the same zones of the surface of the Earth with different resolutions of each cell. This affects not only to the number of cells of the image but the digital levels that represents the height of the DEM.

Following the description in the previous paragraph, the methodology is divided into two phases:

1. Preprocessing: This phase is designed to determine general differences in scale, position, and rotation (rigid transformation).
2. Determination of homologous points (keypoints): This is the main process of the proposed method. It determines the points with the same characteristics in both DEMs using an increasing and rotating ring and a set of pixels surrounding the

selected one with a known Manhattan distance for comparison inside a predefined search window.

The preprocessing phase, due to low variance, is achieved using an Iterative Closest Point (ICP) algorithm following the ideas of Fitzgibbon [22] and Sahillioglu and Kavan [23], among others. The last one was used to achieve a difference in the pixel spatial resolution, as was indicated in previous paragraphs. The points used to carry out this alignment between the two DEMs were a set of local maxima and minima with the following criteria: (i) the maximum or minimum must have a distance higher than the diagonal of the image divided by the number of the total local maxima and minima points detected; (ii) maxima and minimum must be absolute inside a defined zone; and (iii) maximum and minimum must be separated to obtain a match. With this approach, we first obtain a transformation to determine the original differences in scale, rotation, and position. We think that this approach is the most interesting because it isolates our proposed algorithm from knowledge about spatial information. However, a general transformation can be performed due to geoposition parameters of each DEM, which can allow us to test the results.

Once the preprocessing phase is finished, we determine a more precise match inside a defined area around a cell of the reference image projected into the tested image. The improvement in the keypoint position is achieved using the transformation parameters of the preprocessing phase (Figure 2a).

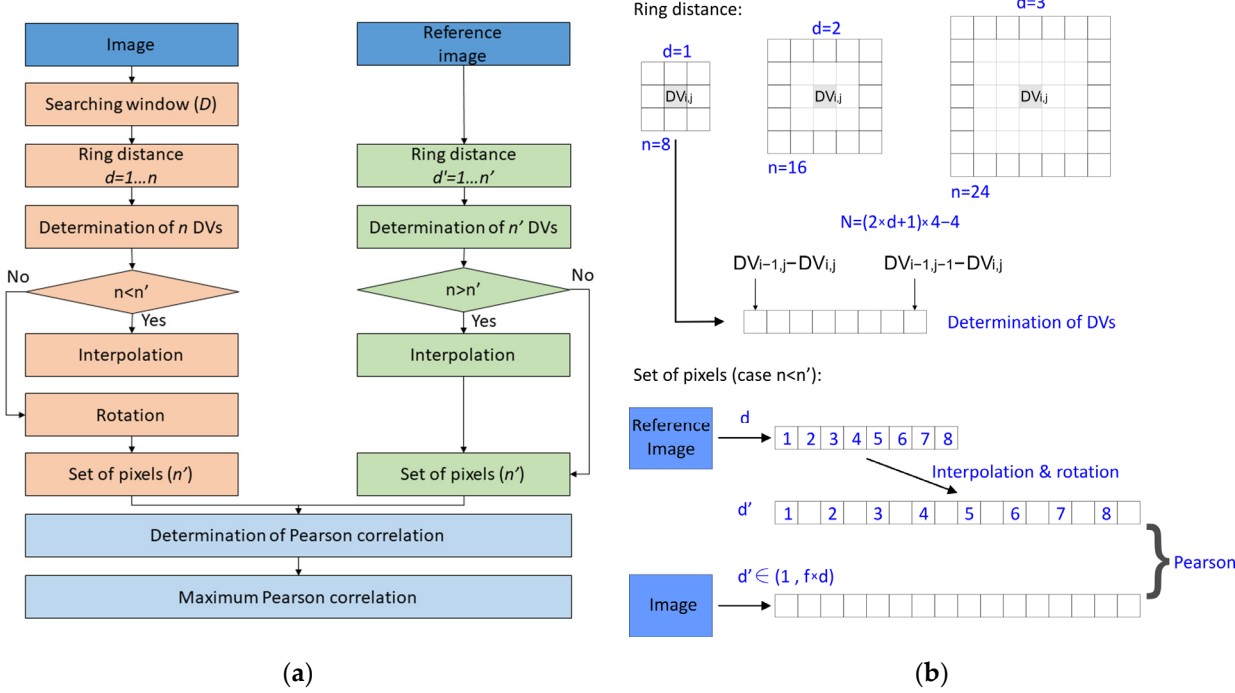

(**a**)                (**b**)

**Figure 2.** Methodology proposed in this study: (**a**) workflow; (**b**) example of the procedure for obtaining two sets of pixels to correlate. The acronyms are as follows: DV: Digital value of pixel; n: number of cells in the ring; d: distance in pixels from the center of the ring in the reference image; D: distance of the search window in pixels in the search image; n: number of pixels of the ring of distance d; n': number of pixels of the ring of distance d'; f: relation between approximate pixel spatial resolution from reference image and search image.

The main idea behind the matching is the use of Pearson chi-squared test value to determine if two rings are similar "enough" to be considered the same. Because this test is defined for two sets of data with the same length, as we consider a set of discrete cells, we can compare both rings. In addition, if we define a rotation direction of one of the rings and we match the number of cells of each ring by means of a simple interpolation (Figure 2b), rings with different orientations and different resolutions per pixel can be compared.

The proposed procedure is described as Algorithm 1. It is a general iterator in the reference image over the tested image using the transformation obtained in preprocessing and a zone around this position. Then, each ring inside this zone is tested against the original ring of the reference image using a rotation. From all of these rings, the maximum correlation coefficient is retrieved. Finally, if the value is greater than a chosen threshold, this match is selected.

| Algorithm 1. Determine matched keypoints between two images. | |
| --- | --- |
| Input | ref_img: Image (Image with one band no restriction about datatype)<br>img: Image (Image with one band no restriction about datatype)<br>d: Distance of the ring in cell units<br>t: Transformation (General transformation from ref_img to img that has to be reversible)<br>s: Relation of scale between ref_img and img<br>minp: Minimum value of Pearson test to consider a keypoint |
| Output | l: List of matched keypoints |
| Pseudocode | l = list()<br>**for each** cell(i,j) **in** img1<br>ring1 = getRing(i,j,d,ref_img)<br>i2, j2 = transform(i,j,t)<br>pbase, rotationbase = 0, 0<br>ibase, jbase = i2, j2<br>**for each** cell(ir2,jr2) **in** range([i2 − d × s,i2 + d × s], [j2 −d × s,j2 + d × s])<br>ring2 = getRing(ir2,jr2,img)<br>ring2scale = scale(ring2, len(ring1))<br>p = pearson(ring1,ring2scale)<br>r = 0<br>**for each** rotation **in** ring_perimeter<br>ring2rotated = rotate(ring2scale, rotation)<br>protated = pearson(ring1, ring2rotated)<br>**if** p < protated **then**<br>p = protated<br>r = rotation<br>**if** pbase < p **then**<br>ibase, jbase = ir2,jr2<br>pbase, rotationbase = p, r<br>**if** pbase >= minp **then**<br>l.add((i,j),(ibase,jbase),(pbase,rotationbase))<br>**return** l |

With regard to the theorical time, it is close to order $O(n^4)$. However, some improvements can be applied, for example, because the rotation of the ring, which is a linear element, is only a translation in the offset of the initial index of the element. On the other hand, the rings can be read in a cached way in the tested image side because the comparison of each ring is applied various times, one for each pixel of the reference image inside the possible matching zone. With these two changes, the theorical time can be reduced. Moreover, if we consider that correlation test has several sums that are not affected by rotation, only the crossed products between the ring from the tested image and the reference image are calculated in the order defined by rotation.

The final approach of the proposed algorithm is to recalculate the initial transformation and determine the differences between transformed keypoints from reference image and their homologous points in tested image. These differences are considered local changes, while the transformation is considered the positional quality.

Finally, it is important to note that there are some parameters that define the goodness of the keypoints' definition. First, the comparison ring distance, because of its low variance, can be defined by several rings. The other parameter is the minimum value of correlation test; in this case, a very high value greater than 0.9 is recommended, because of the low

variance of DEM data. With regard to the other parameters, like scale between DEMs or the test zone maximum search distance, they are defined using the preprocessing phase, so they have fixed default values. However, maximum search distance can be modified if both DEMs are extremely different.

## 3. Description of Datasets

The datasets selected to test the proposed methodology were composed of three types of DEMs related to four cartographic sheets of about 30 km × 19 km (Figure 3). These 12 DEMs are produced and published by the Instituto Geográfico Nacional of Spain [24]. The main properties of these datasets are shown in Table 1, including spatial resolution, dimensions, maximum and minimum heights, and mean slope. Thus, we selected two DTMs with different spatial resolutions (25 and 5 m) and one DSM (spatial resolution of 5 m) from each zone. The selection of the zones was related to their relief. Our aim was to cover all possible cases from flat to mountainous areas (one flat: 1034; two intermedium: 946 and 1008; one mountainous: 1027). In this sense, we selected zones characterized by low to high elevations and slopes (Table 1). We have to note that the difference in spatial resolution (25 vs. 5 m) supposes a small reduction in the height variability and mean slope.

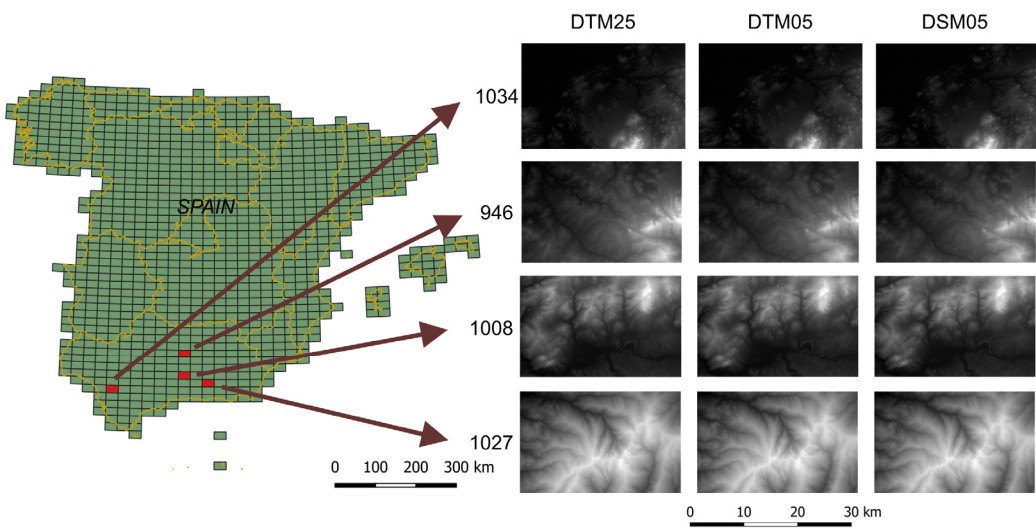

**Figure 3.** DEMs used in the application of the proposed methodology.

**Table 1.** Main characteristics of the DEMs used in this study.

| DEM | Zone | Resolution (m) | Rows/Columns | Min./Max. (m) | Mean Slope (%) | DEM-ID |
|---|---|---|---|---|---|---|
| DTM25 | | 25 | 1189/759 | 271/1307 | 14 | DTM25-0946 |
| DTM05 | 946 | 5 | 5937/3785 | 266/1414 | 15 | DTM05-0946 |
| DSM05 | | 5 | 5935/3783 | 266/1417 | 18 | DSM05-0946 |
| DTM25 | | 25 | 1197/758 | 422/1591 | 19 | DTM25-1008 |
| DTM05 | 1008 | 5 | 5975/3785 | 422/1589 | 21 | DTM05-1008 |
| DSM05 | | 5 | 5973/3783 | 419/1589 | 25 | DSM05-1008 |
| DTM25 | | 25 | 1194/750 | 741/3454 | 43 | DTM25-1027 |
| DTM05 | 1027 | 5 | 5963/3743 | 745/3455 | 46 | DTM05-1027 |
| DSM05 | | 5 | 5960/3741 | 746/3455 | 48 | DSM05-1027 |
| DTM25 | | 25 | 1218/784 | −1/408 | 5 | DTM25-1034 |
| DTM05 | 1034 | 5 | 6081/3911 | −2/406 | 6 | DTM05-1034 |
| DSM05 | | 5 | 6079/3908 | −2/405 | 8 | DSM05-1034 |

The selection of datasets considering two different spatial resolutions (DETM25 and DTM05), representing the same terrain surface, focused on contrasting the capacity of our

method in these cases and determining the influence of this basic aspect of gridded data. In this sense, we selected a differential factor of five between their resolutions, which is a high value considering the purpose of quality assessments. In addition, we also wanted to analyze the feasibility of our approach to detect homologous points when the represented surfaces include important differences. For instance, the DTM05 uniquely represents the surface of the terrain while the DSM05 also includes other elements such as vegetation, constructions, etc. These elements could hide possible terrain points to be used, reducing the number of detected points.

## 4. Results and Discussion

The results section is structured in two parts. The first section shows the results of applying a standard algorithm to obtain the homologous points, SIFT, in each DEM. The second part shows the results obtained by the algorithm proposed in this paper.

### 4.1. Results of SIFT

In order to test the improvements to our methodology to determine the homologous points in two DEMs versus standard keypoint matching algorithms, we propose the use of the SIFT algorithm [15]. Because the algorithm is extensively described and programmed in several environments, we have applied the algorithm using the OpenCV [25]. The results are shown in Table 2 using the standard values proposed by Lowe [15], without limits in the number of feature detections (octaves = 3; contrast threshold = 0.03; sigma of Gaussian = 1.6; edge threshold = 10). In addition, Table 3 shows the matching results obtained from the SIFT BFMatcher algorithm from OpenCV [25]. First, it is important to note that not all keypoints can be used in the search of homologous pixels between two DEMs of the same zone. With respect to the results, the comparison between DTM25 and DSM05 or DTM05 has a scale near 0.2 that reflects the difference in resolution of the pixel indicated in Section 3, while the comparison between DSM05 and DTM05 has a scale similar to 1. In addition, the precision of the transformation is good when the number of the match is low, which could represent a great dispersion in the keypoints' results and distribution.

**Table 2.** Number of keypoints obtained from SIFT algorithm.

| DEM-ID | #Keypoints |
|---|---|
| DTM25-0946 | 11 |
| DTM05-0946 | 9 |
| DSM05-0946 | 8 |
| DTM25-1008 | 40 |
| DTM05-1008 | 45 |
| DSM05-1008 | 41 |
| DTM25-1027 | 27 |
| DTM05-1027 | 27 |
| DSM05-1027 | 27 |
| DTM25-1034 | 22 |
| DTM05-1034 | 24 |
| DSM05-1034 | 23 |

**Table 3.** Matching results of the SIFT algorithm ($\hat{\sigma}_0^2$ is the variance of the mean square adjustment between both DEMS).

| DEM-ID 1 | DEM-ID 2 | #Keypoints for Matching | Mean Scale (X/Y) | $\hat{\sigma}_0^2$ |
|---|---|---|---|---|
| DSM05-0946 | DTM05-0946 | 6 | 1.002052461 | 9.185842889 |
| DSM05-0946 | DTM25-0946 | 8 | 0.200578805 | 0.406204984 |
| DTM05-0946 | DTM25-0946 | 8 | 0.200655345 | 0.257664539 |
| DSM05-1008 | DTM05-1008 | 40 | 0.999798052 | 43.72951535 |
| DSM05-1008 | DTM25-1008 | 29 | 0.173301694 | 40.81601968 |
| DTM05-1008 | DTM25-1008 | 33 | 0.182126235 | 995.761413 |
| DSM05-1027 | DTM05-1027 | 27 | 0.999976911 | 1.831542813 |
| DSM05-1027 | DTM25-1027 | 18 | 0.190169276 | 2007.481743 |
| DTM05-1027 | DTM25-1027 | 18 | 0.190137543 | 1996.775669 |
| DSM05-1034 | DTM05-1034 | 20 | 0.999792179 | 3.298975751 |
| DSM05-1034 | DTM25-1034 | 18 | 0.201131872 | 636.0208014 |
| DTM05-1034 | DTM25-1034 | 19 | 0.200028957 | 0.061698366 |

*4.2. Results of the Proposed Algorithm*

With regard to the results of the algorithm, it was tested on the DEMs presented in Section 3. Moreover, the parameters used were limited to a distance of 1 and a minimum correlation of 0.9995. Following these restrictions, the results are shown in Figures 4–6 and Table 4.

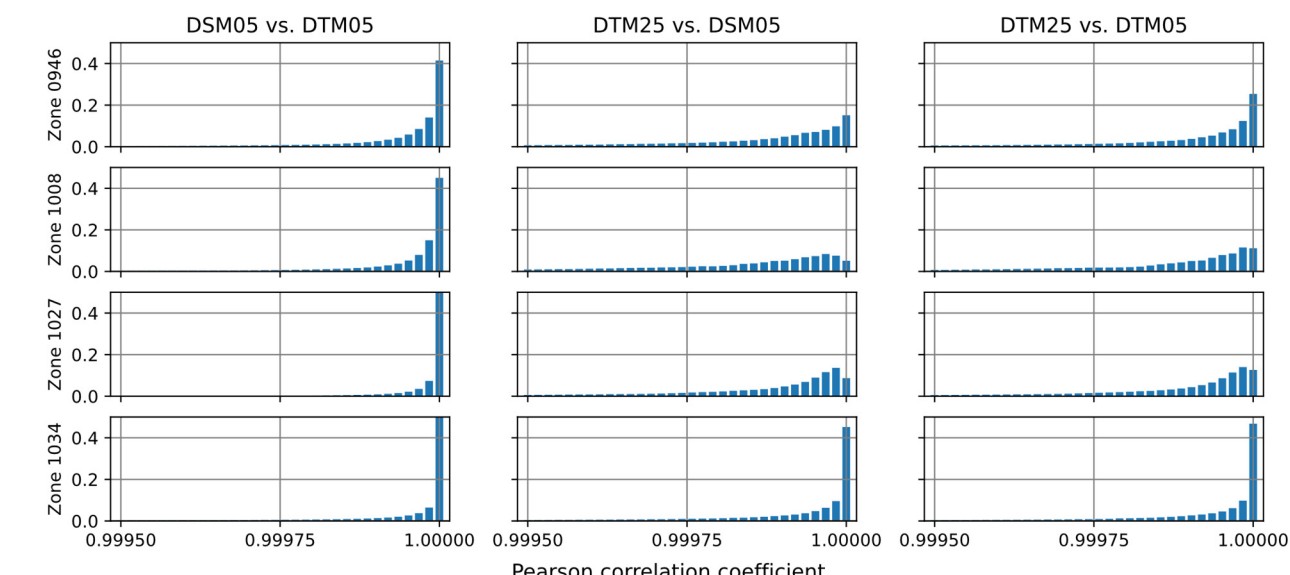

**Figure 4.** Frequency of correlation coefficient values for all the zones and all comparisons.

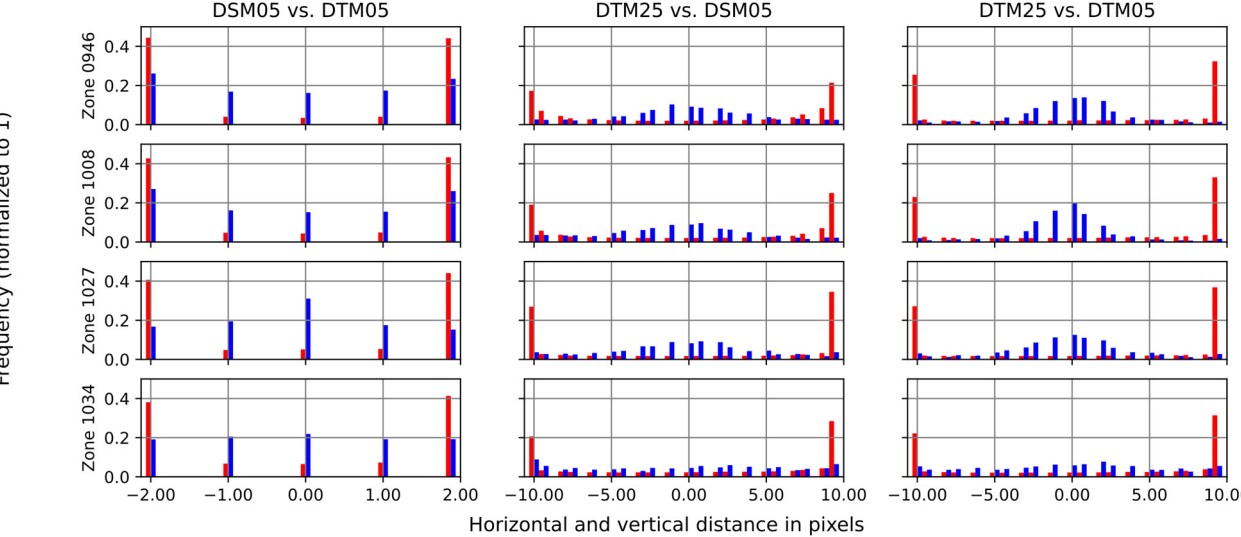

**Figure 5.** Distances in rows and columns of the matched keypoints (red: vertical; blue: horizontal).

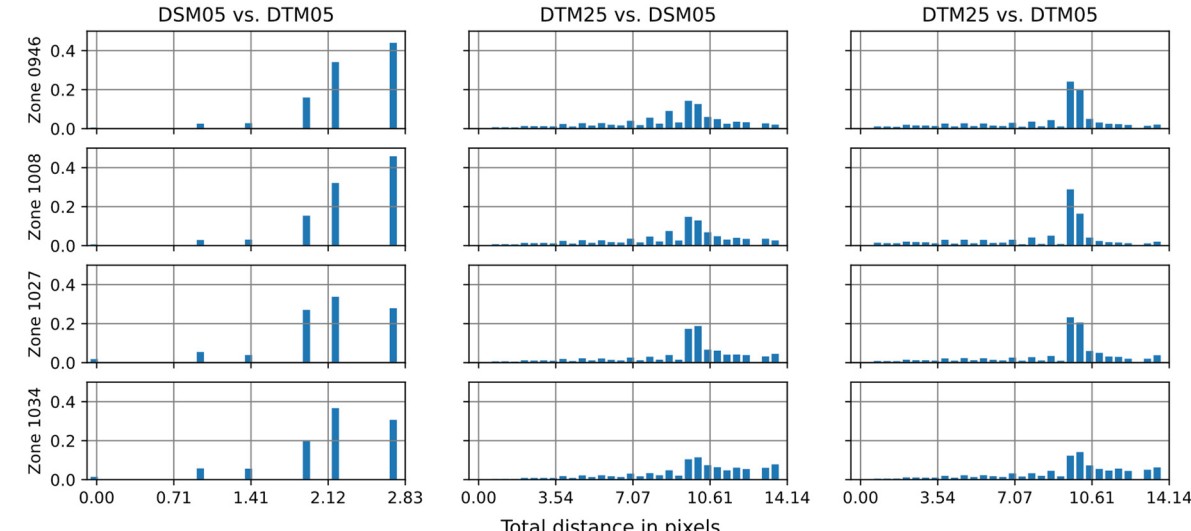

**Figure 6.** Total distance of matched keypoints.

**Table 4.** Matching results of the proposed algorithm ($\hat{\sigma}_0^2$ is the variance of the mean square adjustment between both DEMS).

| DEM-ID 1 (Reference) | DEM-ID 2 | Mean Scale (X/Y) | Rotation | $\hat{\sigma}_0^2$ |
|---|---|---|---|---|
| DSM05-0946 | DTM05-0946 | 1.00000771 | −0.000273299 | 2.9712019 |
| DTM25-0946 | DSM05-0946 | 4.99989885 | 0.000460667 | 43.7015771 |
| DTM25-0946 | DTM05-0946 | 4.99976919 | −0.001847645 | 42.6738983 |
| DSM05-1008 | DTM05-1008 | 1.00001888 | −0.000348352 | 2.98831609 |
| DTM25-1008 | DSM05-1008 | 4.99759005 | −0.000261853 | 44.5912646 |
| DTM25-1008 | DTM05-1008 | 4.99858267 | −0.00105328 | 40.2658607 |
| DSM05-1027 | DTM05-1027 | 0.99997725 | $-4.98485 \times 10^{-5}$ | 2.56474212 |
| DTM25-1027 | DSM05-1027 | 4.99907199 | −0.01462987 | 49.0939488 |
| DTM25-1027 | DTM05-1027 | 4.99895135 | −0.016072889 | 46.0422685 |
| DSM05-1034 | DTM05-1034 | 0.99999601 | 0.000655466 | 2.6222421 |
| DTM25-1034 | DSM05-1034 | 4.99897657 | 0.00246365 | 52.7526375 |
| DTM25-1034 | DTM05-1034 | 4.99930689 | 0.009161375 | 50.3271811 |

After reviewing the correlation values (see Figure 4) a great number of pixels that matched between both DEMs were observed; in fact, the majority of pixels from the DEM with the smaller size were obtained, and the number was always greater than 80%. In addition, most of the pixels were close to 1, except for zones 1008 and 1027 between DEMs of different resolutions.

On the other hand, Figure 5 shows the pixel distance along each axis with different colors for the pixels in the image (not the reference image). A first look reveals that the match between DTM05 and DSM05 only had a maximum distance of 2 pixels, but the distance between DTM25 and the others was up to 10 pixels. These values are defined by the initial scale from the first transformation. Additionally, the graphics indicate a difference between each axis. Following this, the blue axis (horizontal) is closer to a normal distribution with zero, which should mean that the displacement of the matching in this direction is near to zero as well. However, the red axis (vertical) has a different distribution; in this case, the majority of the matched pixels are within the limits of the search window. For this reason, there is supposed to be a great displacement between the matched DEMs on this axis.

Finally, we composed both displacements at a distance expressed in the pixels of the image. The rotation between rings was not considered due to their values being zero in almost all cases, which is explained by the fact that all the DEMs have the same orientation. The composed distance is featured in Figure 6, which presents two different results. The comparison between DSM05 and DTM05 has similar distance values, between two and three pixels. However, between DTM25 and the other DEMs, the distance is around ten pixels, which is two multiplied by the difference in scale of the DEMs. For this reason, both comparisons show a similar result.

With regard to the specific cells that were matched between the DEMs, the images in Figure 7 can be used as an example. In this figure, the red pixels represent cells that have a correlation coefficient better than 0.99. The first idea that we want to highlight is the difference in the matched cells; zone 946 has a smaller number of highlighted cells because there are great differences between MDT and MDS, even with the same spatial resolution. These differences are the expected results because height changes are focused on zones with vegetation and buildings. On the other hand, zone 1027 has a high number of matched cells, which we consider a good result because the DEMs are in areas with low vegetation or even bare ground. The results in Figure 7 also highlight riverside vegetation, for example, in the zoomed areas of zones 1027 and 1034. Moreover, vegetation areas, like the zoomed part of zone 1008, have been detected and discarded as highlighted cells for matching both DEMs.

To end this section, the results of the matching process are shown in Table 4. As we can see, the matched keypoints show almost no rotation and a scale that is almost equal to the scale of definition of the DEMs, which are one between DTM05 and DSM05 and five between DTM25 and the other two. The number of keypoints used for this calculation is more than 500,000 points in the lower cases, and the precision is near the distance of Figure 6 for transformations between DTM05 and DSM05 and five times higher between DTM25 and DSM05 or DTM05.

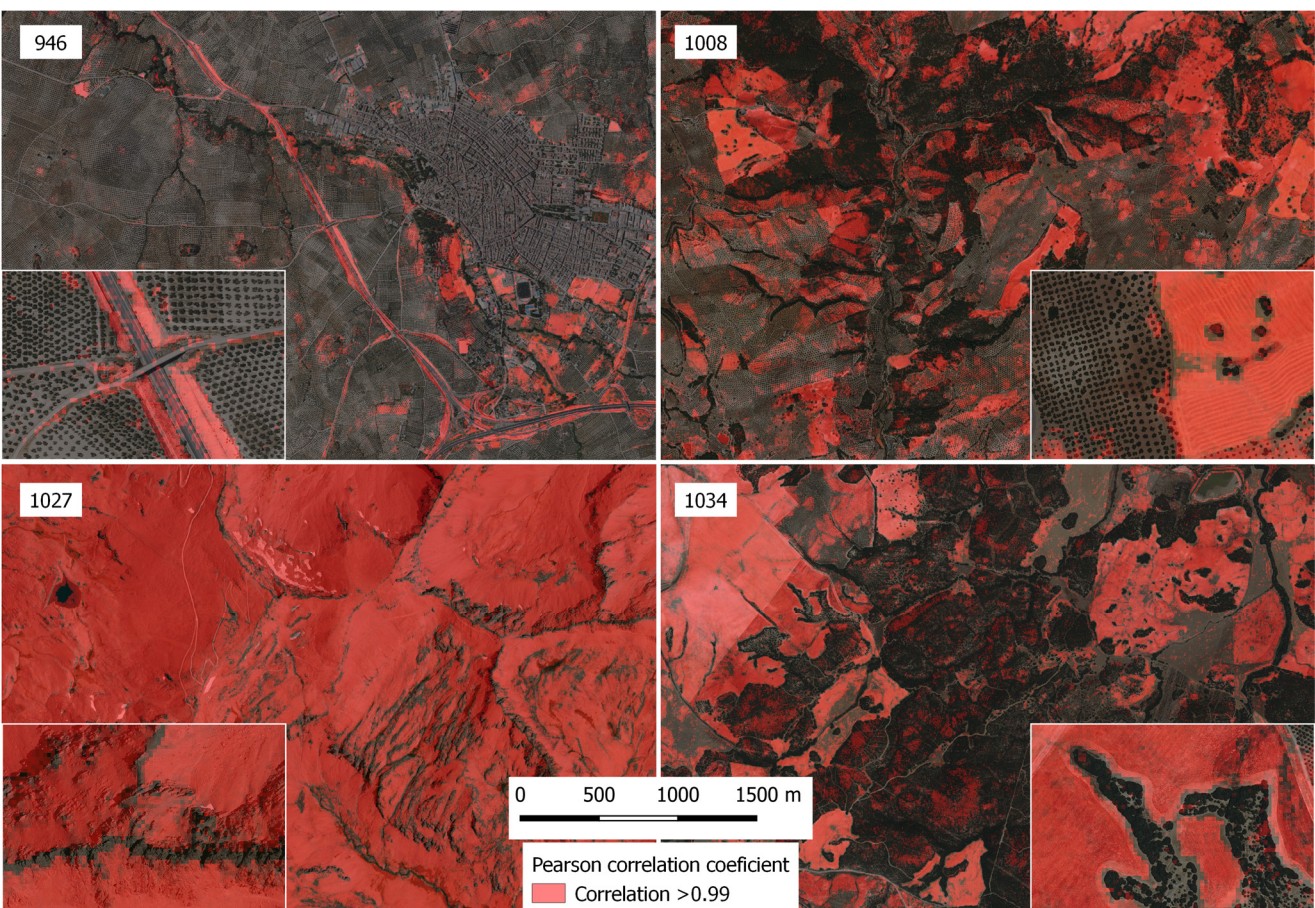

**Figure 7.** Examples of correlation coefficients greater than 0.99 (red colored zones) for the four tested zones. The DEMs were DTM05 compared with DSM05. The background is the orthophoto map of Spain (IGN of Spain) [24].

*4.3. Discussion*

In view of the results shown in this study in Sections 4.1 and 4.2, we could state that the proposed algorithm has some advantages and a few drawbacks with respect to standard feature extraction algorithms like SIFT [15] in the case of DEMs.

First, our approach has demonstrated great potential for obtaining large sets of keypoints (homologous points) from two DEMs with low variance. In this sense, the proposed algorithm allows us to determine a more robust matching and transformation between two images.

Second, we have proved that the algorithm is capable of matching two DEMs regardless of their differences in resolution and geographical areas. This has been achieved by the use of a two-step process: first, a general ICP alignment [22,23], and then, a fine alignment by means of the keypoints. The final alignment allows us to use a variable-sized ring that can be used to analyze different levels of generalization for DEMs, as we have shown between DTM05 and DTM25.

Third, the large set of keypoints determined by the proposed algorithm allows us to obtain a general transformation between DEMs to define a global quality assessment. In addition, we can analyze the local differences among matched keypoints to determine small changes between DEMs. This will allow us to detect changes in zones for multitemporal studies.

Finally, the set of keypoints will be used in all the quality control assessment standards, like ASPRS [12], allowing us to fit the sample requirements of this standard. Moreover, the value of the Pearson correlation of each keypoint match can be used to weigh or filter the

points of the desired sample. Furthermore, keypoint matching can allow us to obtain a horizontal quality control assessment for DEMs.

On the other hand, the algorithm has a few drawbacks. We highlight that the proposed methodology is time-consuming, especially in the fine matching procedure. Even using the improvements indicted in the methodology sections, it takes several hours to obtain the correlation in the worst cases (MDT05 vs. MDS05), using one threading processing under Python 3.10. For this reason, we have reimplemented using PyOpenCL 1.2 and tile overlapping in a i7-8550U CPU with an integrated GPU UHD 620. With this implementation, using the 24 computing units for GPU processing, the algorithm has been boosted and is able to be used to compare the two larger DEMS (MDS05 vs. MDT05) in a maximum of 300 s.

The second drawback that we detected is related to the necessity of a preprocessing alignment between the DEMs. However, the last procedure can be avoided in cases of well-defined DEMs, which include a transformation between the geographical and image coordinates. In this case, the fine matching can help determine the quality of the transformation.

## 5. Conclusions

This study has presented a new algorithm to obtain and match keypoints in low-variance images like DEMs. Our results have demonstrated the feasibility of matching DEMs with different resolutions, different variances, and a large set of keypoints. This will allow us to apply different quality control assessments to large samples. Moreover, the large sets of matched keypoints can determine the changing zones by displacement. In addition, the zones with missing keypoints could define great changes between the DEMs. Both types of information can be used for multitemporal studies.

The algorithm has been tested using large sets of data (DEMs), with differences in variance, resolutions, slopes, and other height descriptors, which has confirmed the viability of the proposed methodology. Moreover, all the used data are publicly available and can be checked by any other researcher.

Our future work will focus on improving the initial alignment using other keypoints with enrichment descriptors of pixels surpassing the proposed local maxima and minima and checking the different approaches for testing, like multirings and other zone matching descriptors that will be defined using simulations based on known deformations to test the accuracy of these changes.

**Author Contributions:** Conceptualization, M.A.U.-C. and A.T.M.-C.; methodology, M.A.U.-C. and A.T.M.-C.; software, M.A.U.-C.; validation, M.A.U.-C.; formal analysis, M.A.U.-C.; investigation, M.A.U.-C. and A.T.M.-C.; resources, M.A.U.-C. and A.T.M.-C.; data curation, M.A.U.-C.; writing—original draft preparation, M.A.U.-C. and A.T.M.-C.; writing—review and editing, M.A.U.-C. and A.T.M.-C.; visualization, M.A.U.-C. and A.T.M.-C.; supervision, M.A.U.-C. and A.T.M.-C.; project administration, M.A.U.-C. and A.T.M.-C.; funding acquisition, M.A.U.-C. and A.T.M.-C. All authors have read and agreed to the published version of the manuscript.

**Funding:** This research received no external funding.

**Data Availability Statement:** All DEMs data is available at IGN Spain website [24].

**Conflicts of Interest:** The authors declare no conflict of interest.

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
