# Peer review of "Identification of Highlighted Cells in Low-Variance Raster Data Application to Digital Elevation Models"

_algorithms, doi:10.3390/a16060302_

Round 1

Reviewer 1 Report

The subject taken up by the authors is important and certainly this study has a chance to be noticed and used by other researchers. My reservations are raised, on the one hand, by linguistic and editorial errors and ambiguities in the description of the methodology, and, on the other hand, by a too limited and one-sided description of testing its effectiveness. Comments on editorial and linguistic errors have been added to the text of the article in the attached PDF file.

The authors will pay attention to the time-consuming calculations performed with the algorithm proposed by them. I am asking for specific information on this subject in relation to the tested datasets. In which programming language was the code used in the calculations written? Did the code use multithreading? What hardware was used for the calculations?

Section 4.2, which contains a description of the results obtained using the tested algorithm, is limited to very basic information, which does not give a complete picture of the matter. Firstly, there is no data on the local variation in the density of keypoints and the factors that may affect it. In one sentence, the authors indicate that the sensitivity of the algorithm in this area may be useful for determining small differences in DEM, especially in the case of analyzing changes over time. It would also make sense to experiment more extensively with the DEMs being tested, for example by introducing globally or locally random errors (deviations) and rotating them.

Author Response

Reviewer #1:

Authors: Thank you for your suggestions and comments. We have rewritten the paper following your suggestions.

Reviewer #1:

Open Review

( ) I would not like to sign my review report

(x) I would like to sign my review report

Quality of English Language

( ) I am not qualified to assess the quality of English in this paper

( ) English very difficult to understand/incomprehensible

( ) Extensive editing of English language required

(x) Moderate editing of English language required

( ) Minor editing of English language required

( ) English language fine. No issues detected

Yes      Can be improved        Must be improved       Not applicable

Does the introduction provide sufficient background and include all relevant references?

(x)       ( )        ( )        ( )

Are all the cited references relevant to the research?

(x)       ( )        ( )        ( )

Is the research design appropriate?

( )        (x)       ( )        ( )

Are the methods adequately described?

( )        (x)       ( )        ( )

Are the results clearly presented?

( )        ( )        (x)       ( )

Are the conclusions supported by the results?

( )        ( )        (x)       ( )

Comments and Suggestions for Authors

The subject taken up by the authors is important and certainly this study has a chance to be noticed and used by other researchers. My reservations are raised, on the one hand, by linguistic and editorial errors and ambiguities in the description of the methodology, and, on the other hand, by a too limited and one-sided description of testing its effectiveness. Comments on editorial and linguistic errors have been added to the text of the article in the attached PDF file.

Authors: Thank you for your suggestions. We have improved the manuscript following yours comments on PDF.

Reviewer #1: The authors will pay attention to the time-consuming calculations performed with the algorithm proposed by them. I am asking for specific information on this subject in relation to the tested datasets. In which programming language was the code used in the calculations written? Did the code use multithreading? What hardware was used for the calculations?

Authors: We have included a description of the programming language, used hardware and time spent on calculations. Also, we have finished the GPU programming and included this part in the paper. We also have removed the reference on “Future works”. Thank you

Reviewer #1: Section 4.2, which contains a description of the results obtained using the tested algorithm, is limited to very basic information, which does not give a complete picture of the matter. Firstly, there is no data on the local variation in the density of keypoints and the factors that may affect it. In one sentence, the authors indicate that the sensitivity of the algorithm in this area may be useful for determining small differences in DEM, especially in the case of analyzing changes over time. It would also make sense to experiment more extensively with the DEMs being tested, for example by introducing globally or locally random errors (deviations) and rotating them.

Authors: We have attended your suggestion including Figure 7 with represents local variations of matched keypoints. In addition we have briefly analyzed the matched keypoints results in relation to the ground represented by DEMs. Thank you.

Reviewer 2 Report

This work proposes a keypoint matching technique for DEM models. Writing and organization are acceptable. I suggest improving the performance analysis part - compare your method against the state-of-the-art.

1.How do you obtain the reference DEM?

2. What is(are) the difference(s) between the reference image and the test image?

3. How do you make sure the refence DEM is accurate enough?

4. Line 176, 202 - Explain "ring"

5. Line 271 - "both DEMs" - be specific

6. For table 3, what was(were) the reference DEM(s)? DEM-ID1 or DEM-ID2 or the combinations shown in Fig 4?

 7. Fig 4 -  What is your interpretation for Fig 4?

8. Figs 4,5- As you are comparing your method against the SIFT or SURF, it would be interesting to see the same performance analysis for the SIFT/SURF keypoint matching (or a suitable state-of-the-art method) and then compare it against your algorithm.

Minor editing of English language required

Author Response

Reviewer #2:

Authors: Thank you for your suggestions and comments. We have rewritten the paper following your suggestions.

Reviewer #2:

Open Review

(x) I would not like to sign my review report

( ) I would like to sign my review report

Quality of English Language

( ) I am not qualified to assess the quality of English in this paper

( ) English very difficult to understand/incomprehensible

( ) Extensive editing of English language required

( ) Moderate editing of English language required

(x) Minor editing of English language required

( ) English language fine. No issues detected

Yes      Can be improved        Must be improved       Not applicable

Does the introduction provide sufficient background and include all relevant references?

(x)       ( )        ( )        ( )

Are all the cited references relevant to the research?

(x)       ( )        ( )        ( )

Is the research design appropriate?

( )        (x)       ( )        ( )

Are the methods adequately described?

( )        (x)       ( )        ( )

Are the results clearly presented?

( )        ( )        (x)       ( )

Are the conclusions supported by the results?

( )        (x)       ( )        ( )

Comments and Suggestions for Authors

This work proposes a keypoint matching technique for DEM models. Writing and organization are acceptable. I suggest improving the performance analysis part - compare your method against the state-of-the-art.

1.How do you obtain the reference DEM?

Authors: All DEMs are obtained from an official cartographic institution in Spain (IGN). So we have used one of them as a reference based on the minimum resolution.

Reviewer #2: 2. What is(are) the difference(s) between the reference image and the test image?

Authors: The differences between images are spatial resolution (DTM05 and DTM25) and differences of the height definition, one is the terrain height without vegetation and buildings while the other DEM included these height information (DTM05 and DSM05). This have been described in the text.

Reviewer #2: 3. How do you make sure the refence DEM is accurate enough?

Authors: It is not necessary because we are not determining absolute controls. Our approach is focused on obtaining matching between both DEMs. However, our algorithm can be used to analyze relative changes.

Reviewer #2: 4. Line 176, 202 - Explain "ring"

Authors: Done.

Reviewer #2: 5. Line 271 - "both DEMs" - be specific

Authors: Thank you. We have changed this sentence for improving the meaning.

Reviewer #2: 6. For table 3, what was(were) the reference DEM(s)? DEM-ID1 or DEM-ID2 or the combinations shown in Fig 4?

Authors: Done.

Reviewer #2: 7. Fig 4 -  What is your interpretation for Fig 4?

Authors: As it is described in the text, Figure 4 show that there is a great number of matched pixels using a high value of correlation coefficient (0.99). We also have included a new Figure showing local results and in the Discussion section we also describe this high number of keypoints matched.

Reviewer #2: 8. Figs 4,5- As you are comparing your method against the SIFT or SURF, it would be interesting to see the same performance analysis for the SIFT/SURF keypoint matching (or a suitable state-of-the-art method) and then compare it against your algorithm.

Authors: We think that the number of SIFT keypoints is extremely low to be shown in a Figure like 4 or 5. For this reason we have included the number of SIFT keypoints in Table 2 and the matched keypoints in Table 3. Thank you.

Round 2

Reviewer 1 Report

The authors of the reviewed article in its second version introduced most of the postulated changes and additions, and clarified earlier doubts. I found only two or three places that require minor correction (comments added to the text in the attached PDF file).

My only doubt concerns the representativeness of the DEM/DSM models used to test the proposed algorithm. A broader geological and geomorphological description of selected sheets would be needed. In my opinion, the main criterion should be the scale of relief variability in relation to DEM resolution, which is often related to its genesis. In my country, for example, apart from mountains and highlands of various ages and geological structures, most of the territory is occupied by lowlands with relief shaped during the Pleistocene glaciations. Depending on the age of the glaciation, this relief differs in vertical scale and horizontal extent. On some types of relief (for example, dune fields or terminal moraines) the resolution of 25 m is too small to capture first-order forms.

Author Response

Reviewer #1:

Open Review

( ) I would not like to sign my review report

(x) I would like to sign my review report

Quality of English Language

( ) I am not qualified to assess the quality of English in this paper

( ) English very difficult to understand/incomprehensible

( ) Extensive editing of English language required

( ) Moderate editing of English language required

(x) Minor editing of English language required

( ) English language fine. No issues detected

Yes         Can be improved            Must be improved         Not applicable

Does the introduction provide sufficient background and include all relevant references?

(x)          ( )           ( )           ( )

Are all the cited references relevant to the research?

(x)          ( )           ( )           ( )

Is the research design appropriate?

( )           (x)          ( )           ( )

Are the methods adequately described?

(x)          ( )           ( )           ( )

Are the results clearly presented?

( )           (x)          ( )           ( )

Are the conclusions supported by the results?

( )           (x)          ( )           ( )

Comments and Suggestions for Authors

The authors of the reviewed article in its second version introduced most of the postulated changes and additions, and clarified earlier doubts. I found only two or three places that require minor correction (comments added to the text in the attached PDF file).

Authors: We have improved the text considering your suggestions.

Reviewer #1:

My only doubt concerns the representativeness of the DEM/DSM models used to test the proposed algorithm. A broader geological and geomorphological description of selected sheets would be needed. In my opinion, the main criterion should be the scale of relief variability in relation to DEM resolution, which is often related to its genesis. In my country, for example, apart from mountains and highlands of various ages and geological structures, most of the territory is occupied by lowlands with relief shaped during the Pleistocene glaciations. Depending on the age of the glaciation, this relief differs in vertical scale and horizontal extent. On some types of relief (for example, dune fields or terminal moraines) the resolution of 25 m is too small to capture first-order forms.

Authors: As you said, we agree that certain geomorphological structures are only represented on DEMs considering a specific spatial resolution. However, our procedure is not concern of the geomorphology of terrain. In addition our approach only tries to detect homologous points between two DEMs.  We use all relief forms regardless the geological genesis. We used four areas (one flat, two intermedium and one mountainous) (see table 1) to check differences of variance of height. Moreover, the difference of spatial resolution (25 vs 5 meters) supposes a small reduction in height variability and mean slope. We have improved the text considering your suggestion. Thank you.

Reviewer 2 Report

Authors have addressed my comments.

Minor editing of English language required

Author Response

Reviewer #2:

Open Review

(x) I would not like to sign my review report

( ) I would like to sign my review report

Quality of English Language

( ) I am not qualified to assess the quality of English in this paper

( ) English very difficult to understand/incomprehensible

( ) Extensive editing of English language required

( ) Moderate editing of English language required

(x) Minor editing of English language required

( ) English language fine. No issues detected

Yes         Can be improved            Must be improved         Not applicable

Does the introduction provide sufficient background and include all relevant references?

(x)          ( )           ( )           ( )

Are all the cited references relevant to the research?

(x)          ( )           ( )           ( )

Is the research design appropriate?

( )           (x)          ( )           ( )

Are the methods adequately described?

(x)          ( )           ( )           ( )

Are the results clearly presented?

(x)          ( )           ( )           ( )

Are the conclusions supported by the results?

(x)          ( )           ( )           ( )

Comments and Suggestions for Authors

Authors have addressed my comments.

Comments on the Quality of English Language

Minor editing of English language required

Authors: Thank you. We have improved the text considering your suggestion.